# Visual Function and Survival of Injured Retinal Ganglion Cells in Aged Rbfox1 Knockout Animals

**DOI:** 10.3390/cells11213401

**Published:** 2022-10-27

**Authors:** Lei Gu, Jacky M. K. Kwong, Joseph Caprioli, Natik Piri

**Affiliations:** 1Stein Eye Institute, University of California, Los Angeles, CA 90095, USA; 2Brain Research Institute, University of California, Los Angeles, CA 90095, USA

**Keywords:** retina, ganglion cells, amacrine cells, optic nerve crush, depth perception

## Abstract

Rbfox1 is a multifunctional RNA binding protein that regulates various aspects of RNA metabolism important for neuronal differentiation and normal physiology. Rbfox1 has been associated with neurodevelopmental and neurological conditions as well as age-related neurodegenerative diseases such as Alzheimer’s and Parkinson’s. We have shown that in mammalian retinas Rbfox1 is expressed in retinal ganglion cells (RGCs) and in amacrine cells (ACs). This study investigates the effect of advanced age (22-month-old mice) on visual function, retinal morphology and survival of injured retinal ganglion cells (RGC) in Rbfox1 knockout (KO) animals. A visual cliff test, which was used to evaluate visual function, showed that 22-month old Rbfox1 KO mice have profound depth perception deficiency. Retinal gross morphology in these animals appeared to be normal. Optic nerve crush (ONC) induced axonal injury resulted in approximately 50% of RGC loss in both Rbfox1 KO and age-matched control animals: the average RGC densities in uninjured control and Rbfox1 KO animals were 6274 ± 1673 cells/mm^2^ and 6004 ± 1531 cells/mm^2^, respectively, whereas 1 week after ONC, RGC numbers in the retinas of control and Rbfox1 KO mice were reduced to 2998 ± 858 cells/mm^2^ and 3036 ± 857 cells/mm^2^, respectively (Rbfox1 KO vs. Rbfox1 KO + ONC, *p* < 0.0001 and control vs. control + ONC, *p* < 0.0001). No significant difference between RGC numbers in Rbfox1 KO + ONC and age-matched control + ONC animals was observed, suggesting that Rbfox1 has no effect on the survival of injured RGCs. Interestingly, however, contrary to a commonly accepted view that the number of RGCs in old (18 month of age) compared to young animals is reduced by approximately 40%, the RGC densities in 22-month-old mice in this study were similar to those of 4-month-old counterparts.

## 1. Introduction

Rbfox1 is a one of three members of the Rbfox family of multifunctional RNA binding proteins that are involved in regulation of RNA metabolism, including alternative splicing, transcription, mRNA stability and translation [1,2]. The diverse functions of these protein are supported by spatiotemporal expression of different isoforms generated by alternative splicing or from alternative promoters. Rbfox proteins, Rbfox1, Rbfox2 and Rbfox3, are expressed in neurons, heart and skeletal muscle, and disruption of their normal functions have been associated with neurodevelopmental and neuropsychiatric disorders, including autism spectrum disorder (ASD), intellectual disability, epilepsy, ADHD, bipolar disorder, schizoaffective disorder and schizophrenia, sleep latency and heart disease [3,4,5,6,7,8,9]. The expression of Rbfox genes overlap in most areas of the brain, but their spatial pattern of expression in the cerebellar cortex, for instance, is quite different: granule cells express Rbfox1 and Rbfox3, whereas Purkinje cells express Rbfox1 and Rbfox2 [10,11]. The Rbfox proteins also exhibit different subcellular localization. Rbfox1 expression is observed in both the cytoplasm and nucleus of Purkinje cells, whereas Rbfox2 is restricted to the nucleus. The Rbfox genes have distinct patterns of expression also during cerebellar development.

In the retina, expression of Rbfox1, Rbfox2 and Rbfox3 overlaps in retinal ganglion cells (RGCs) and amacrine cells (ACs) [12,13,14,15]. However, there are notable differences: Rbfox2 and Rbfox3, but not Rbfox1, are expressed in horizontal cells (HCs) and Rbfox2 is expressed in a wider population of ACs and displaced ACs (dACs) subtypes within the inner nuclear layer (INL) and ganglion cell layer (GCL), respectively, than Rbfox1. Furthermore, although both Rbfox1 and Rbfox2 are expressed throughout retinal development; Rbfox1 expression shifts from cytoplasmic to predominantly nuclear at around P0 and remains so in mature retinas, whereas Rbfox2 localization is predominantly nuclear during retinogenesis and in adult retinas [15]. These differences in spatial and temporal expression suggest specific roles of Rbfox paralogs during retinal neurogenesis and in maintaining normal physiology of differentiated neurons. Transcriptome analysis identified multiple genes involved in synaptic functions that are regulated by Rbfox1 and Rbfox2 as well genes associated with circadian rhythm/entrainment pathways regulated by Rbfox2. Interestingly, a single gene deletion of Rbfox1, Rbfox2 or Rbfox3 in adult animals had no effect on gross retinal morphology, although both Rbfox1^−/−^ and Rbfox2^−/−^ mice experienced depth perception deficiency, which indicates the role of these proteins in regulation of gene networks associated with the retino-geniculo-cortical pathway [12,14,15]. Since all three Rbfox paralogs recognize the same (U)GCAUG element within their target genes, the absence of cellular phenotype in single Rbfox gene knockout (KO) or knockdown models can be explained by functional redundancy of Rbfox proteins in cells that have overlapping expression of these genes. On the other hand, individual Rbfox proteins have specific roles during neuronal development and in mature neuron physiology [10,11,16,17,18,19,20,21,22] and deficiencies in different Rbfox paralogs have been associated with the various neurodevelopmental and neurological conditions mentioned above. Recent studies also implicate Rbfox proteins, particularly Rbfox1, in the pathogenesis of age-related neurodegenerative diseases such as Alzheimer’s and Parkinson’s diseases [23,24,25,26,27,28,29]. Based on these observations, we can speculate that the effect of Rbfox1 functional deficiency on normal cellular physiology and integrity may be influenced by additional genetic/epigenetic factors, including those that are associated with age- and stress-related changes. Earlier, we showed that downregulation of Rbfox1 in young adult mice (4 months old) has no effect on the number of surviving RGCs after ONC injury [30]. The current study evaluates the effect of advanced age and optic nerve injury on retinal morphology, visual function and the survival of RGCs in Rbfox1 KO animals.

## 2. Experimental Procedures

### 2.1. Generation of Rbfox1 KO Animals

Experimental procedures with animals were approved by the Animal Research Committee of the University of California at Los Angeles and were performed in compliance with the National Institutes of Health Guide for the Care and Use of Animals and the ARVO (The Association for Research in Vision and Ophthalmology) Statement for the Use of Animals in Ophthalmic and Vision Research. Animals were housed in a 12 h light-dark cycle with food and water available ad libitum. Rbfox1 KO animals were generated by crossing homozygous transgenic Rbfox1^fl/fl^ mice (loxP sites flanking Rbfox1 gene exons 11–12; kindly provided by Dr. Douglas Black, UCLA; [11]) with Tg(UBC-Cre/ERT2)1Ejb mice (Jackson Laboratory, Bar Harbor, ME). The UBC-Cre-ERT2 transgenic mouse line, in which the expression of tamoxifen-inducible Cre recombinase (Cre) gene is controlled by the human ubiquitin C (UBC) promoter, was used in this study since it provides a strong expression of Cre in RGCs (www.informatics.jax.org/recombinase/specificity?id=MGI:3707333&system=sensory+organs, accessed on 20 August 2022; [15,31]). The resulting heterozygous Rbfox1^fl/+^/UBC-Cre^+/−^ animals were bred with Rbfox1^fl/fl^ mice to generate homozygous Rbfox1^fl/fl^/UBC-Cre^+/−^ mice. Cre activity in homozygous Rbfox1^fl/fl^/UBC-Cre^+/−^ animals was induced with tamoxifen (Sigma, St. Louis, MO, USA). Rbfox1^fl/fl^/UBC-Cre^+/−^ and age-matched heterozygous Rbfox1^fl/+^ control mice were administered 5 doses of tamoxifen solution (200 mg/kg; once a day) by oral gavage. Rbfox1 KO animals were viable, had a normal growth rate and behaviorally had no apparent anomalies compared to control animals. Age-matched heterozygous Rbfox1^fl/+^ mice were used as controls.

### 2.2. Visual Cliff Test

Visual function in Rbfox1 KO animals was examined by evaluation of depth perception with cliff test. The protocol described earlier was followed [14,15,32,33]. The test uses a glass-bottomed box placed on the edge of the table so its one half is resting on the table surface (“shallow” side) and the other half is suspended above the floor (the “cliff drop” or “deep” side; Figure 1A). The illusion of the cliff was created by covering the table and the floor under the test box with a black and white checkered tablecloth. The time the animal spent on the “shallow” and “deep” sides during 5 min of testing was recorded. The vibrissae of the tested mice were removed so the tactile placing responses do not interfere with evaluation of visual function. The glass surface of the test box was thoroughly cleaned after each test. Six Rbfox1 KO and six control animals were tested. Five independent experiments for each animal were performed.

### 2.3. Optic Nerve Crush (ONC) Injury

Prior to the ONC, external ocular examination was performed on all the anesthetized mice. Adnexal (the eyelids and conjunctiva), and anterior segment structures (cornea, iris and sclera) were examined by slit lamp biomicroscopy at ×16 magnification using broad-beam illumination. ONC was performed as described previously with minor modifications (Nadal-Nicolás et al., 2009). Briefly, the optic nerve was exposed and crushed with fine forceps (#5 Dumont, Fine Science Tools, Foster City, CA, USA) approximately 1 mm behind the posterior pole of the eye for 1 sec. Care was taken not to damage the blood vessels. The procedure was performed unilaterally.

### 2.4. Inclusion and Exclusion Criteria

Twenty-two-month-old Rbfox1^fl/fl^/UBC-Cre^+/−^ and age-matched heterozygous Rbfox1^fl/+^ control mice were included in this study. Prior to the ONC procedure, exclusion criteria were visible corneal abrasion, opacity, inflammation or edema and cataract. A fundus examination was performed under an operating microscope immediately after the ONC procedure to exclude animals with blockage of retinal blood flow from the study. The exclusion criteria were also included: surgical complications including infection, bleeding, lens discoloration, or purulent drainage from the wound. No animal that entered the study was rejected due to the above-listed exclusion criteria.

### 2.5. Retinal Sections and Immunohistochemistry

Eyes were enucleated, fixed with ice-cold 4% paraformaldehyde and cryoprotected in 30% sucrose. 14-µm thick retinal sections were cut with cryostat. For immunohistochemistry, sections were incubated with blocking solution (20% fetal calf serum, 5% goat serum, 0.1% Triton X-100 in PBS) for 30 min and then with primary antibodies at 4 °C overnight. Primary antibodies used in this study: anti-Rbfox1 produced in mouse (1:200; Novus Biologicals, Littleton, CO, USA), anti-Rbpms produced in guinea pig (1:1000; PhosphoSolutions, Aurora, CO, USA; [34]) and anti-calbindin D-28K produced in rabbit (1:500; EMD Millipore, Billerica, MA). These primary antibodies have been well characterized and have high specificity for their target protein. After incubation with the primary antibodies, sections were washed with 0.1% Triton X-100 in PBS and incubated with secondary antibodies for 1 h at room temperature. Secondary antibodies used in this study: Alexa Fluor 488-conjugated goat anti-rabbit IgG, Alexa Fluor 568-conjugated goat anti-mouse IgG and Alexa Fluor 488-conjugated goat-anti-guinea pig IgG (all 1:500; Thermo Fisher Scientific, Canoga Park, CA, USA). Sections were mounted with mounting medium containing DAPI for nuclear counterstaining and imaged with a confocal laser scanning microscope Olympus FV3000 (Olympus FV3000, Tokyo, Japan). 

### 2.6. Cell Quantification in Whole Mount Retinas

A standard protocol was followed for cell quantification in retinal flat mounts [34]. Briefly, the enucleated eyeballs were fixed in 4% paraformaldehyde in 0.1 M phosphate buffer, the retinas were dissected and incubated with 10% serum for 1 h to reduce nonspecific staining and then with anti-Rbfox1 produced in mouse (1:200; Novus Biologicals) or anti-Rbpms produced in guinea pig (1:1000; PhosphoSolutions) primary antibodies overnight at 4 °C. Retinas were washed and then incubated with the corresponding secondary antibody overnight at 4 °C. Secondary antibodies used in this experiment: Alexa Fluor 568-conjugated donkey anti-mouse IgG and Alexa Fluor 488-conjugated goat-anti-guinea pig IgG (all 1:500; Thermo Fisher Scientific). The retinas were placed flat with the GCL facing upward, divided into superior, inferior, nasal and temporal quadrants with several radial cuts and mounted flat on the glass slide. Four sampling fields per retinal quadrant (0.24 × 0.24 mm each) were imaged at 0.5, 1, 1.5 and 2 mm from the center of the optic nerve disc with a confocal laser scanning microscopy (Olympus FV3000). Retinas from four animals per group were used in these experiments. Quantification was performed in a masked manner; the person who counted the cells had no knowledge about samples used for the quantitative analysis.

### 2.7. Statistical Analysis

An unpaired Student’s *t*-test was used for analysis of the visual cliff test data and cell density data. *p* < 0.05 was considered statistically significant. Data are presented as mean ± SD.

## 3. Results

### 3.1. Depth Perception Deficiency in Old Rbfox1 KO Animals

To evaluate visual function in aged Rbfox1 null mice, a visual cliff test was used (Figure 1A). Animals with normal vision have the innate tendency to avoid the “cliff” field and stay on the “shallow” side for most of the testing time. Twenty-two-month-old Rbfox1 KO mice and age-matched control heterozygote Rbfox1^fl/+^ animals were evaluated. Downregulation of Rbfox1 was completed one month prior to the test. Visual function evaluation in this test is based on the time the animals spend in the “deep” versus “shallow” side of the field. Control mice, as expected, showed a strong preference for the “shallow” side of the field. The mean (+/−SD) time that the control group (*n* = 5) spent on the “deep” side was 36.6 +/− 25.3 s during 300 s of testing time (Figure 1B). Rbfox1 KO animals, unlike their control counterparts, were indifferent to either side of the test field and spent almost as much time on the “deep” side as on the “shallow” side: the mean time that the Rbfox1 KO animals (*n* = 5) spent on the “deep” side was 141.8 +/− 64.6 s during 300 s of testing time (Figure 1B). The mean difference in time spent on the “deep” side between two groups was statistically significant: 105.2 s (*p* < 0.0001).

### 3.2. The Effect of Aging and ONC-Induced Injury on Retinal Morphology in Rbfox1 KO Animals

Four groups of aged (twenty-two-month-old) animals were evaluated: uninjured control heterozygote Rbfox1^fl/+^ mice, control mice injured by ONC, Rbfox1 KO uninjured and Rbfox1 KO injured by ONC. Tamoxifen-induced downregulation of Rbfox1 was completed one month and ONC was performed 1 week before animals were euthanized for histological evaluation. In the uninjured control group, the expression of Rbfox1 was observed in cells located in the GCL and in cells within the innermost layer of the INL (Figure 2A). In the GCL, these Rbfox1-positive cells were also stained with the RGC marker, Rbpms, or with calbindin, a neuronal marker, which was used to label ACs (most ACs in the INL and dACs in the GCL express calbindin). Virtually all Rbpms positive cells (RGCs) and the vast majority of calbindin positive cells in the GCL (dACs and a small number of RGCs express calbindin) in these aged animals were Rbfox1-positive (Figure 2A,B). In the INL, most ACs adjacent to the inner plexiform layer (IPL) were Rbfox1-positive (Figure 2B). Calbindin-stained HCs (pointed with white arrows) in the outermost INL were Rbfox1-negative. ONC resulted in significant reduction in RGCs: very few Rbfox1/Rbpms-positive cells were present, but the majority of Rbfox1 immunostained cells in the GCL were calbindin-positive, suggesting that most of them are dACs (Figure 2C,D). ONC had no effect on intensity and pattern of Rbfox1 expression in the INL. Quantitative analysis of ONC-induced cell loss is presented in “The effect of aging on RGC numbers and their survival after axonal injury in Rbfox1 KO animals” section below.

In Rbfox1 KO animals, a dramatic reduction of Rbfox1 immunostaining intensity within the GCL and INL was observed (Figure 3A–D). Very few faintly stained for Rbfox1 RGCs and ACs were present. The immunostaining pattern or intensity for Rbpms and calbindin was not affected by Rbfox1 downregulation (Figure 3A,B). The retinal gross morphology in these animals was normal and similar to that of the control mice described above. An extensive loss of Rbpms-labeled RGCs was observed 1 week after ONC (Figure 3C). The staining intensity for Rbpms in remaining RGCs was much weaker compared to that of uninjured cells (Figure 3A,C). Interestingly, in both control (Figure 2B,D) and Rbfox1 KO (Figure 3B,D) animals, the intensity of calbindin immunostaining 1 week after ONC was also visibly reduced in the GCL, IPL, INL and especially in the outer plexiform layer (OPL) where photoreceptors make synaptic connections with bipolar cells and HCs.

### 3.3. The Effect of Aging on RGC Numbers and Their Survival after Axonal Injury in Rbfox1 KO Animals

To determine the effect of aging on the number of RGCs in Rbfox1 KO mice and the rate of their survival after ONC-induced axonal injury, Rbpms- and Rbpms/Rbfox1-labeled cells were counted in whole mounted retinas. As for immunohistochemical analysis, for groups of twenty-two-month-old animals were used: uninjured and ONC-injured control mice and uninjured and ONC-injured Rbfox1 KO mice. Representative images of retinas from each of these groups that were immunostained for Rbpms and Rbfox1 are shown in Figure 4. Images representing ONC retinas (Figure 4B,D) show a significant reduction in Rbpms labeled cells compared to corresponding uninjured controls (Figure 4A,C). Reduction of Rbfox1-labeld cells in retinas of control mice injured by ONC is due to the loss of RGCs (Figure 4B), whereas the loss of Rbfox1 immunoreactivity in Rbfox1 KO transgenes is the result of Rbfox1 downregulation (Figure 4C) or a combination of Rbfox1 downregulation and ONC-induced loss of RGCs (Figure 4D). Quantitative data for Rbpms- and Rbpms/Rbfox1-positive cells that were counted in superior, inferior, nasal and temporal retinal quadrants at 0.5, 1, 1.5 and 2 mm from the center of the optic nerve disc are presented in Figure 5 and Figure 6 and Table 1 and Table 2. The number of Rbpms-positive cells in the retinas of uninjured control and Rbfox1 KO mice were very similar when compared at corresponding locations (Figure 5A, Table 1). The average densities were: 6273 ± 1673 and 6004 ± 1531 Rbpms-positive cells/mm^2^ for control and Rbfox1 KO mice, respectively (*n* = 4/group; Figure 5B). One week after ONC, the numbers of Rbpms-stained cells in retinas of control and Rbfox1 KO mice were decreased to 2998 ± 858 cells/mm^2^ and 3036 ± 857 cells/mm^2^, respectively (*n* = 4/group; Figure 5B). These numbers indicate approximately 50% RGC loss caused by ONC compared to uninjured animals (control vs. control + ONC, *p* = 2.75 × 10^−27^ and Rbfox1 KO vs. Rbfox1 KO + ONC, *p* = 2.55 × 10^−26^).

Downregulation of Rbfox1 in Rbfox1 KO mice retinas is not uniform and there are cells both ACs and RGCs, in which Rbfox1 expression is preserved (Figure 3A,B). This raises a question as to whether there is a difference in survival rate of Rbfox1-positive and Rbfox1-negative RGCs after ONC. To address this question, we counted the number of Rbfox1/Rbpms-immunostained cells in the control and Rbfox1 KO groups of animals with and without ONC injury (*n* = 4/group). The data indicate a significant reduction of Rbfox1/Rbpms-positive cells in all four retinal quadrants of Rbfox1 KO mice compared to the control (Figure 6A,B and Table 2). The average densities of Rbfox1/Rbpms-immunostained cells in control and Rbfox1 KO retinas were 5911 ± 1557 cells/mm^2^ and 1062 ± 634 cells/mm^2^, respectively, which translates to more than 80% of Rbfox1-negative RGCs in Rbfox1 KO animals (*p* = 4.50 × 10^−47^; Figure 6B). ONC injury resulted in the reduction of Rbfox1/Rbpms-positive cells by ~ 58% in control animals (5911 ± 1557 cells/mm^2^ in control vs. 2215 ± 849 cells/mm^2^ in control + ONC, *p* = 1.14 × 10^−33^) and by ~ 50% in Rbfox1 KO animals (1062 ± 634 cells/mm^2^ in Rbfox1 KO vs. 489 ± 340 cells/mm^2^ in Rbfox1 KO + ONC, *p* = 3.19 × 10^−9^; Figure 6B). Quantitative analysis of Rbfox1/Rbpms-positive Cells shows that Rbfox1 has no significant effect on survival of RGCs injured by ONC.

## 4. Discussion

The present study evaluates the effect of aging on visual function, retinal morphology and survival of RGCs after axonal injury in Rbfox1 KO animals. Twenty-two-month-old Rbfox1 KO and age-matched control mice were used as “old” mice. Mice between 18–24 months of age meet the definition of “old,” which is characterized by the presence of senescent changes in almost all biomarkers in all animals [35]. Significant impairment of cognitive function is present at 22 months of age [36].

We first evaluated the effect of aging on visual function in Rbfox1 KO animals. The visual cliff test that assesses the integrity and function of retino-geniculo-cortical pathway was used. In our earlier work, we have shown that downregulation of Rbfox1 in young adult mice (4 months old) results in deficient depth perception [15]. These animals spent more time on the “deep” side of the test field than on the “shallow” side. The aged Rbfox1 KO animals were also depth perception “blind” and spent almost as much time on the “deep” side as on the “shallow” side. Both young and aged control animals as expected, had a clear preference for the “shallow” side and avoided the illusionary cliff. The Rbfox1 KO model used in this study produced a robust downregulation of Rbfox1 in RGCs. However, downregulation of Rbfox1 in this model is not restricted to RGCs and can take place in other neurons, including those involved in the retino-geniculo-cortical pathway within the lateral geniculate nucleus or visual cortical regions. Therefore, the observed deficiency in depth perception may be associated with altered retinal function or with neuronal dysfunction in brain regions that process this visual information. Even though we cannot pinpoint the exact location at which retino-geniculo-cortical pathway is affected at this time, the results of this test show the importance of Rbfox1 in regulation of genes associated with this visual function.

Retinal gross morphology in aged Rbfox1 KO animals, in which more than 80% of RGCs were Rbfox1 negative, was unremarkable compared to age-matched controls. With respect to the effect of aging on the survival of injured neurons in Rbfox1 KO animals, we anticipated to see: (a) an increased rate of RGC degeneration in aged Rbfox1 KO animals compared to younger animals under similar conditions or (b) an increased loss of RGCs in Rbfox1 KO animals compared to the age-matched control with normal level of Rbfox1 expression. The first assumption is based on a decreased capacity of aged cells to oppose the destabilizing effects of metabolic stressors. Normal neuronal aging can be viewed as a metabolic state characterized by a decreased homeostatic reserve. Cellular homeostasis under normal conditions and in the presence of various stressors is maintained with the help of stress-response signaling pathways, such as the heat shock response (HSR) and the unfolded protein responses of the mitochondria (UPR^MT^) and endoplasmic reticulum (UPR^ER^), all of which are known to deteriorate during aging [37]. Therefore, acute surges of metabolic activity (induced by injury or by a loss of a protein, such as Rbfox1, which regulates the expression of genes important for normal neuronal functions) can reach the limits of homeostatic reserve defenses that are incompatible with the normal levels of functional load and result in an increased age-dependent vulnerability and extensive neuronal death [38]. The second assumption is based on several studies implicating Rbfox1 in the stress-induced regulation of several mechanisms that stimulate cell survival, including miR-132/Rbfox1-mediated mechanisms that promote neuronal survival against amyloid β-peptide (Aβ) and glutamate excitotoxicity in the brain of the mouse model for Alzheimer’s disease [26,39] 

The effect of Rbfox1 downregulation on survival of injured RGCs was evaluated 1 week after ONC, the time point at which the progressive RGC loss is well underway (~50–60% RGC loss). Quantitative analysis of RGC numbers in aged ONC-injured and uninjured Rbfox1 KO animals showed that downregulation of Rbfox1 has no effect on the RGC density compared to age-matched controls. No significant changes were observed in any of four retinal quadrants; the average densities were 6273 cells/mm^2^ and 6004 cells/mm^2^ for control and Rbfox1 KO mice, respectively and 2998 cells/mm^2^ and 3036 cells/mm^2^ for control + ONC and Rbfox1 KO + ONC, respectively. Recall that, in young mice, downregulation of Rbfox1 also had no detectable effect on the number of uninjured and ONC-injured RGCs [15,30]. What is interesting though, is that the number of RGCs in young (4-month-old) and old (22-month-old) control animals were very similar: 6438 cells/mm^2^ in young [30] vs. 6273 cells/mm^2^ in old animals. Progressive decline of RGCs with age has been documented in studies on rodents, primates and humans. RGC quantification indicates that the average rate of RGC loss in mice is 2.3% per month and in rats is 1.5% per month with a total loss of 41% in mice during 18 months of lifespan and 36% in rats during 24 months of lifespan [40,41]. In monkeys and humans RGC soma and axon quantification estimate 1.7% per year and 0.5% per year RGC loss, respectively resulting in the total loss of 44% during 26 years of a monkey lifespan and 38% during 76 years of a human lifespan [40,42,43,44,45,46]. However, there are studies that question the linearity of RGC loss progression during a lifetime and some even report no significant age-related loss of RGCs or their axons in the optic nerve. For instance, RGC loss in C57BL6 mice was found to start between 12 and 15 months of age with a total of 46% reduction by 18 months of age [47]. On the other hand, a comparison of all major retinal neuronal types, including RGCs, in young adult (3–5 months old) and aged (24–28 months old) mice showed no significant change in their number with age [48]. In Brown Norway rats, no axonal loss was observed in animals up to 24 months of age and only approximately 12% loss was reported by 30 months of age [49]. No age-dependent loss of RGC axons was detected in rhesus monkeys [50]. These discrepancies may be explained by a number of considerations used in the experimental design that can ultimately affect the outcomes of the study. These include the size of each age group, sampling techniques (i.e., a part or an entire optic nerve or retina was analyzed) and RGC labeling technique (immunohistochemical or retrograde; retrograde labeling may fail to label RGC somas if the axoplasmic transport is affected by age). Therefore, a standardized approach would be beneficial for a comprehensive evaluation of RGC loss during aging.

Considering the role of Rbfox1 in the regulation of splicing and transcriptional networks important for neurogenesis and neuronal function, and that the disruption of its functions have been associated with several neurodevelopmental, neuropsychiatric and neurodegenerative diseases, it was expected that deletion of this gene in RGCs would have a dramatic effect on their phenotype and survival in response to injury. The absence of cellular phenotype in this model, as well as other single Rbfox gene KO or knockdown models, can be explained by functional redundancy of Rbfox proteins in cells that have overlapping expression of these genes: all three members of the family, Rbfox 1, Rbfox2 and Rbfox3, are expressed in RGCs, have the same recognition site within their target genes, and therefore may substitute each other’s function.

In conclusion, aged Rbfox1 KO mice have deficient depth perception, which suggest the involvement of Rbfox1 in the regulation of gene(s) responsible for normal function of the retino-geniculo-cortical pathway. Retinal morphology in old Rbfox1 KO animals was normal. Axonal injury resulted in approximately 50% RGC loss in both control and Rbfox1 KO animals 1 week after ONC suggesting that Rbfox1 has no significant effect on vulnerability or survival of injured RGCs.

## Figures and Tables

**Figure 1 cells-11-03401-f001:**
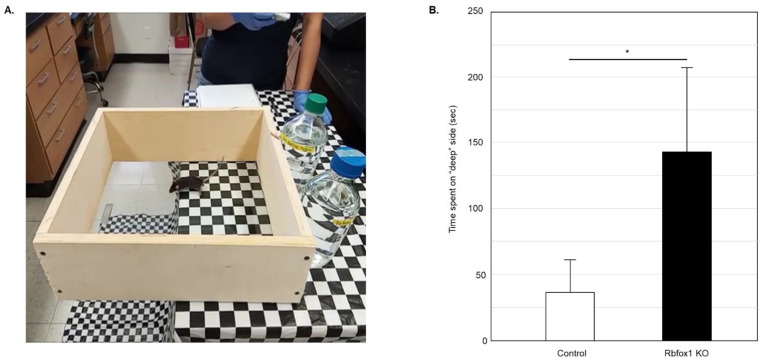
Evaluation of visual function in old Rbfox1 KO mice. (**A**) Visual cliff test setup. A visual cliff illusion was created by placing a glass-bottomed test box on the edge of the table so that one part sits on the table (“shallow” side) and the rest is suspended above the floor (“deep” side). The table and the floor under the test box were covered with black and white checkered table cloth. Animals tested for 5 min and time the spent on the “shallow” and “deep” sides was recorded. Each animal was tested 5 times. (**B**) Control mice spent most of their time on the “shallow” side of the field. The time they spent on the “deep” side was only 36.60 +/− 25.34 s (mean +/− SD; *n* = 5) out of 5 min of testing time. Rbfox1 KO mice showed no preference to either side of the test field. The time the Rbfox1 KO animals spent on the “deep” side was 141.75 +/− 64.63 s (mean +/− SD; *n* = 5). The mean difference in time spent on the “deep” side between two groups was statistically significant: 105.15 s (* *p* = 4.55 × 10^−7^).

**Figure 2 cells-11-03401-f002:**
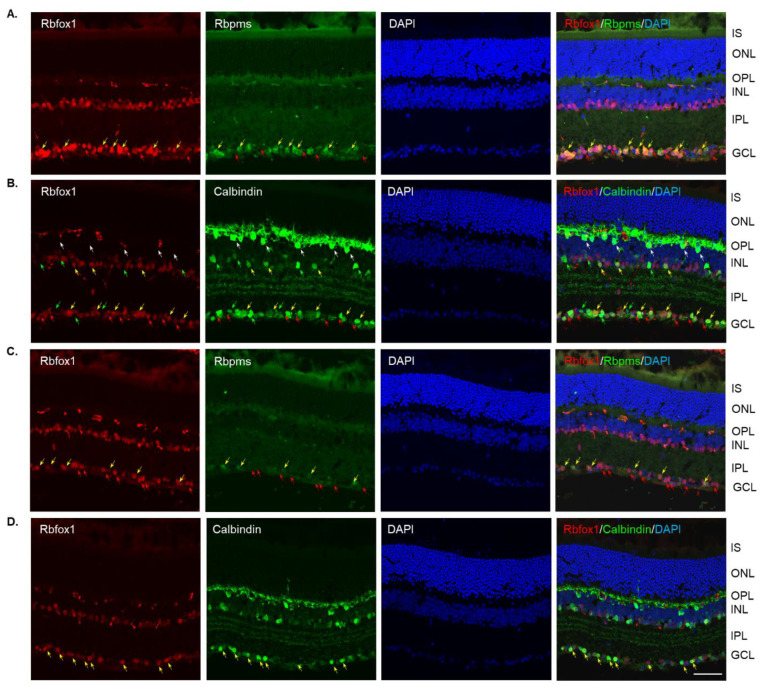
Expression of Rbfox1 in 22-month-old mouse retinas. As in young animals, Rbfox1 expression in the retinas of old mice was restricted to the cells within the GCL, which contains somas of RGCs and dACs in a ratio of approximately 1:1, and the innermost layer of INL, which is occupied by AC somas. (**A**) Virtually all Rbpms-labeled RGCs were also immunostained with Rbfox1 (pointed with yellow arrows). Some dACs in the GCL-Rbfox1-positive/Rbpms-negative cells–are pointed with red arrows. (**B**) Rbfox1 is expressed in the majority of calbindin-positive cells in the GCL (dACs and some RGCs express calbindin) and in the INL (ACs; yellow arrows). Rbfox1-negative dACs and ACs are pointed with green arrows. RGCs (Rbfox1 positive/calbindin negative) are indicated by red arrows. Antibodies against calbindin also stain HCs in the outmost layer of INL (adjacent to the OPL) are pointed with white arrows. (**C**) ONC resulted in significant loss of RGCs. Several remaining Rbpms/Rbfox1-positive cells are indicated by yellow arrows. Red arrows point at Rbfox1 positive dACs. (**D**) Most Rbfox1-positive cells in the GCL were also positive for calbindin, confirming that these cells are dACs (yellow arrows). A decrease in calbindin immunostaining intensity in retinas of ONC mice compared to that of uninjured animals was observed. IS, photoreceptor inner segments; ONL, outer nuclear layer; OPL, outer plexiform layer; INL, inner nuclear layer; IPL, inner plexiform layer; GCL, ganglion cell layer. Scale bar, 50 µm.

**Figure 3 cells-11-03401-f003:**
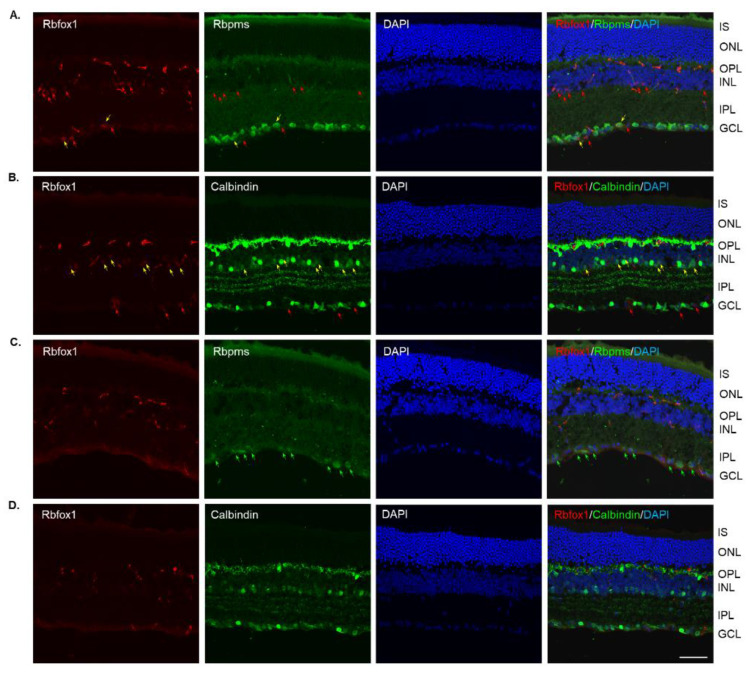
Downregulation of Rbfox1 expression in aged Rbfox1 KO animals. (**A**–**D**). The number of Rbfox1 positive cells and the immunostaining intensity in the few remaining Rbfox1-positive cells were dramatically decreased in retinas of both uninjured and ONC animals. Very few Rbfox1-positive RGCs or ACs were present (**A**,**B**), respectively; yellow arrows). Red arrows point at Rbfox1-immunostained ACs and dACs (**A**) and RGCs (**B**). The number of RGCs, dACs and ACs in Rbfox1 KO animals appeared to be normal. (**C**) Significant loss of RGCs 1 week after ONC was observed. Some of the remaining RGCs are pointed with green arrows. (**D**). The pattern of calbindin immunostaining is unaffected by ONC, although, the staining intensity appears to be lower than that in uninjured retinas. IS, photoreceptor inner segments; ONL, outer nuclear layer; OPL, outer plexiform layer; INL, inner nuclear layer; IPL, inner plexiform layer; GCL, ganglion cell layer. Scale bar, 50 µm.

**Figure 4 cells-11-03401-f004:**
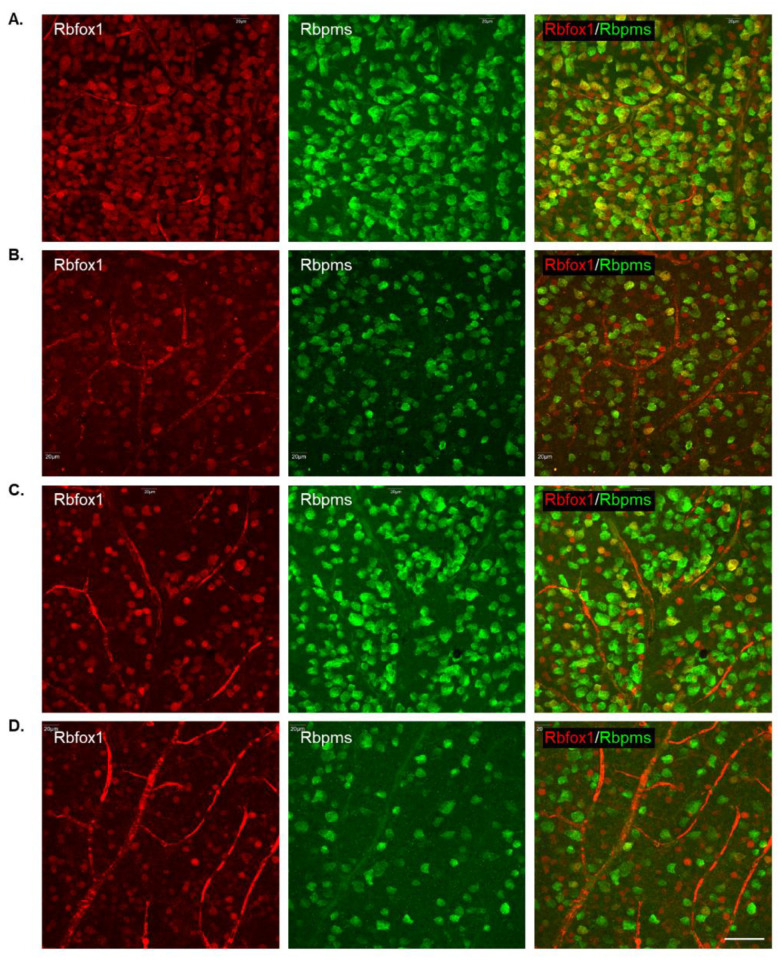
Representative images of immunostained whole-mounted retinas used for quantitative analysis of Rbpms and Rbpms/Rbfox1-positive cells in aged Rbfox1 KO animals with and without ONC injury. For quantitative analysis, Rbpms-positive and Rbpms/Rbfox1-positive cells were counted in the superior, inferior, nasal and temporal retinal quadrants at 0.5, 1, 1.5 and 2 mm from the center of the optic disc. Images represent Rbpms and Rbfox1 immunostained retinas at 1 mm from the center of the optic nerve head (**A**–**D**). A significant reduction in the numbers of Rbpms-labeled cells in ONC groups (**B**,**D**) compared to uninjured animals (**A**,**C**) is evident. Loss of Rbfox1-positive cells in control + ONC (**B**) is due to the loss of injured RGCs. (**A**) dramatic reduction of Rbfox1-positive cells was observed in retinas of Rbfox1 KO animals (**C**). Their numbers were further decreased in Rbfox1 KO + ONC (**D**) due to the loss of RGCs. Scale bar, 50 µm.

**Figure 5 cells-11-03401-f005:**
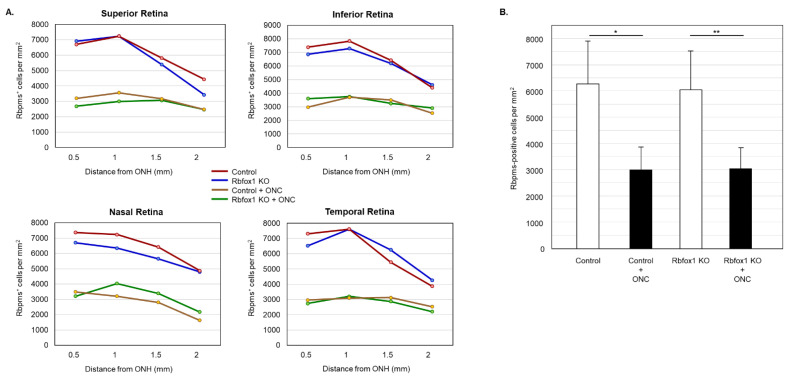
RGC quantification in aged Rbfox1 KO animals with and without ONC. (**A**) The density of Rbpms-positive cells in the superior, inferior, nasal and temporal retinal quadrants at 0.5, 1, and 1.5 mm from the center of the optic disc. (**B**) The number of Rbpms-positive cells in in retinas of uninjured control and Rbfox1 KO mice were very similar when compared at corresponding locations (Figure 5A, Table 1). The average RGC densities in retinas of control and Rbfox1 KO mice were 6273 ± 1673 cells/mm^2^ and 6004 ± 1531 cells/mm^2^, respectively (*n* = 4/group). The average RGC densities in retinas of control and Rbfox1 KO mice one week after ONC were reduced to 2998.32 ± 858.40 cells/mm^2^ and 3036 ± 857 cells/mm^2^, respectively (*n* = 4/group). Thus, ONC resulted in approximately 50% RGC loss compared to uninjured animals (control vs. control + ONC, * *p* = 2.75 × 10^−27^ and Rbfox1 KO vs. Rbfox1 KO + ONC, ** *p* = 2.55 × 10^−26^). No significant difference between RGC densities in control vs. Rbfox1 KO animals was observed. ONC, optic nerve crush; ONH, optic nerve head.

**Figure 6 cells-11-03401-f006:**
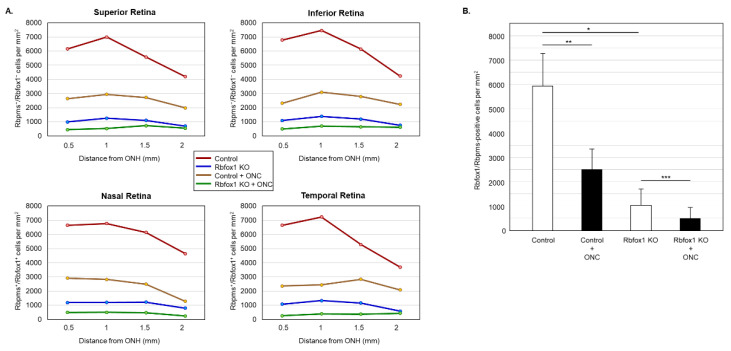
Downregulation of Rbfox1 and its effect on the survival of injured RGCs. For the quantitative assessment of Rbfox1 downregulation within RGC population, the numbers of Rbfox1/Rbpms-immunostained cells in control and Rbfox1 KO animals with and without ONC injury were counted (*n* = 4/group). (**A**) A significant reduction of Rbfox1/Rbpms-positive cells in all four retinal quadrants of Rbfox1 KO mice compared to the control was observed. (**B**) The average densities of Rbfox1/Rbpms-positive cells in control and Rbfox1 KO retinas were 5911 ± 1557 cells/mm^2^ and 1062 ± 634 cells/mm^2^, respectively. This indicates that more than 80% of RGCs in Rbfox1 KO animals are negative for Rbfox1 (* *p* = 4.50 × 10^−47^). ONC-induced loss of Rbfox1/Rbpms-positive cells was approximately 58% in control animals (5911 ± 1557 cells/mm^2^ in control vs. 2215 ± 849 cells/mm^2^ in control + ONC, ** *p* = 1.14 × 10^−33^) and ~50% in Rbfox1 KO animals (1062 ± 634 cells/mm^2^ in Rbfox1 KO vs. 489 ± 340 cells/mm^2^ in Rbfox1 KO + ONC, *** *p* = 3.19 × 10^−9^). ONC, optic nerve crush; ONH, optic nerve head.

**Table 1 cells-11-03401-t001:** Densities of Rbpms-positive cells in retinas of control and Rbfox1 KO animals with and without ONC injury (per mm^2^).

	Location	S 0.5	S 1.0	S 1.5	S 2.0	I 0.5	I 1.0	I 1.5	I 2.0	*n* 0.5	N 1.0	N 1.5	N 2.0	T 0.5	T 1.0	T 1.5	T 2.0
Group																	
Control		6710 ± 1144	7235 ± 780	5824 ± 997	4431 ± 760	7382 ± 803	7826 ± 742	6419 ± 1769	4388 ± 2271	7369 ± 1121	7235 ± 282	6415 ± 1472	4879 ± 1364	7318 ± 1146	7617 ± 901	5447 ± 1635	3872 ± 1337
Control + ONC		3194 ± 605	3563 ± 1010	3168 ± 506	2474 ± 560	2973 ± 823	3702 ± 713	3498 ± 749	2526 ± 475	3490 ± 608	3212 ± 758	2799 ± 1619	1636 ± 322	2969 ± 302	3099 ± 771	3134 ± 956	2535 ± 931
Rbfox1 KO		6918 ± 930	7235 ± 672	5399 ± 1306	3433 ± 1040	6861 ± 509	7282 ± 965	6184 ± 1535	4605 ± 1475	6701 ± 1120	6345 ± 749	5651 ± 1846	4796 ± 1159	6519 ± 262	7604 ± 1170	6258 ± 1265	4266 ± 859
Rbfox1 KO + ONC		2687 ± 530	2995 ± 904	3069 ± 1473	2461 ± 980	3602 ± 975	3750 ± 520	3242 ± 737	2908 ± 530	3212 ± 739	4041 ± 594	3390 ± 887	2183 ± 983	2747 ± 543	3212 ± 698	2873 ± 280	2209 ± 467

Rbpms positive cells (RGCs) were counted in superior (S), inferior (I), nasal (N) and temporal (T) retinal quadrants at 0.5, 1, 1.5 and 2 mm from the center of the optic disc.

**Table 2 cells-11-03401-t002:** Densities of Rbfox1/Rbpms-positive cells in retinas of control and Rbfox1 KO animals with and without ONC (per mm_2_).

	Location	S 0.5	S 1.0	S 1.5	S 2.0	I 0.5	I 1.0	I 1.5	I 2.0	N 0.5	N 1.0	N 1.5	N 2.0	T 0.5	T 1.0	T 1.5	T 2.0
Group																	
Control		6155 ± 1077	6992 ± 688	5577 ± 983	4193 ± 742	6784 ± 428	7465 ± 736	6150 ± 1806	4236 ± 2194	6641 ± 931	6762 ± 278	6137 ± 1445	4635 ± 1361	6641 ± 1108	7231 ± 903	5286 ± 1602	3689 ± 1293
Control + ONC		2633 ± 897	2940 ± 978	2714 ± 797	1979 ± 522	2315 ± 775	3096 ± 1023	2789 ± 959	2228 ± 708	2911 ± 1002	2824 ± 1037	2488 ± 1582	1285 ± 254	2361 ± 437	2436 ± 384	2830 ± 1048	2083 ± 838
Rbfox1 KO		990 ± 580	1246 ± 707	1094 ± 805	690 ± 399	1089 ± 564	1385 ± 669	1185 ± 656	755 ± 386	1194 ± 711	1202 ± 631	1215 ± 834	794 ± 537	1076 ± 629	1328 ± 1103	1155 ± 822	586 ± 395
Rbfox1 KO + ONC		438 ± 186	525 ± 326	725 ± 535	543 ± 420	490 ± 215	690 ± 520	638 ± 486	608 ± 270	490 ± 183	499 ± 429	473 ± 348	239 ± 173	265 ± 142	395 ± 359	369 ± 405	430 ± 388

Rbpms/Rbfox1 positive cells were counted in superior (S), inferior (I), nasal (N) and temporal (T) retinal quadrants at 0.5, 1, 1.5 and 2 mm from the center of the optic disc.

## Data Availability

The raw data supporting the conclusions of this article will be made available by the authors, without undue reservation.

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
