# Peer review of "Visual Function and Survival of Injured Retinal Ganglion Cells in Aged Rbfox1 Knockout Animals"

_cells, 2022, doi:10.3390/cells11213401_

Round 1

Reviewer 1 Report

In their submission, Gu and colleagues attempt to investigate the role of Rbfox1 in visual function and retinal ganglion cell (RGC) susceptibility to degeneration in aged mice. These experiments use a ubiquitous tamoxifen inducible-Cre mouse to knockout Rbfox1 expression in aged mice. The authors report a loss of performance on the visual cliff test in line with their previous publication on non-aged Rbfox1 knockouts. They then show that after optic nerve crush (ONC) RGC loss is similar to control aged mice 1 week after injury. Finally, they demonstrate that although RGC numbers are not changed, the number of Rbfox1 expressing RGCs is lower after ONC than in 'knock-out' controls.

Overall this study suffers from a number of deficiencies.

First, the knockout model is problematic. On the one hand, it is driven by a ubiquitous promoter. As such, Rbfox1 expression in other tissues may contribute to some of the results shown here. For example, this group has previously reported that Rbfox1 is expressed in the cornea. How are the effects on a visual test to be interpreted as a retinal phenotype when the front of the eye could just as likely be altered by Rbfox1 knockout? Rbfox1 knockout may also be altering visual circuits in the brain. A retinal or even RGC specific knockout would be much more powerful.

On the other hand, their delivery of tamoxifen appears to be incomplete and leaves about 1/6th of RGCs and an undocumented number of amacrine and horizontal cells to express Rbfox1. It seems strange that a tamoxifen delivery adequate to achieve complete retinal knockout was not formulated before performing experiments on mice that were aged for nearly two years. This salt and pepper knockout makes the interpretation of the finding that Rbfox1 expressing RGCs are decreased in the knockout rather difficult to interpret and of questionable interest. Although it does seem most likely due to a downregulation in Rbfox1 expression following injury as some reduction is seen in control mice. 

Other important flaws in the experimental design are also present. Calbindin is used as a marker for amacrine cells although it is expressed in some RGCs, and not present in all amacrine cells. AP2 is a more accepted marker of amacrine cells that labels most types and is restricted from RGCs. Additionally, the choice of examining RGC survival 1 week after crush is rather atypical. The majority of studies in the field use 2 weeks as an end point as RGC death reaches a bit over 80% by this point and further loss is significantly slowed thereafter.

Taken together this study demonstrates limited advancement of the field and contains multiple issues of foundational design.

Author Response

We thank the reviewer for the comments and helpful suggestions to improve our manuscript. All of the issues that were brought up were taken into consideration in the revised manuscript and addressed point-by-point below. We believe that the changes made to the manuscript significantly improved its quality.

The inducible knockout model used in this study provided inactivation of the target gene in more than 80% of RGCs. To our knowledge, there is no available cre line that can provide an efficient downregulation of gene expression in all retinal cells, including RGCs, or specifically in RGCs. Although the cre gene is controlled by the ubiquitin promoter, cre expression is not ubiquitous, and, in the retina, its primarily localized to cells in the ganglion cell layer (please see www.informatics.jax.org/recombinase/specificity?id=MGI:3707333&system=sensory+organs). With respect to possible involvement of the cornea in depth perception deficiency, external ocular examination was performed on all mice. Adnexal (the eyelids and conjunctiva), and anterior segment structures (cornea, iris and sclera) were examined by slit lamp biomicroscopy at ×16 magnification using broad-beam illumination. Furthermore, young ~ 2-month-old Rbfox1 KO animals, which also had altered depth perception, were subjected to fundus examination, AC-OCT and SD-OCT. No change, including in corneal transparency, corneal thickness or curvature was noted (unpublished data; were obtained after publication of 2018 Plos1 paper that described these mice). Therefore, we are confident that the visual abnormality in Rbfox1 KO mice are not caused by possible corneal alteration. Regarding the involvement of visual circuits in the brain, we do not exclude this possibility and clearly stated this in 2018 Plos1 paper when described this phenomenon. For this reason, we call it a deficiency in visual function and not a “retinal” phenotype. Even though we can’t pinpoint at this time the exact location at which visual pathway that is involved in depth perception is affected, our observation shows the importance of Rbfox1 in regulation of genes that process this visual information. The manuscript was revised to address these concerns.

We don’t think that the Rbfox1 positive cells in knockout animals were present due to “incomplete“ tamoxifen delivery. Tamoxifen was administered by oral gavage and the possibility that one cell received it and the neighboring cell did not, is very unlikely. In our opinion, incomplete gene knockout is due to selective expression of cre from the ubiquitin promoter. As we stated above, in the retina, the cre expression in this transgene is predominantly localized to the cells within GCL. That is the reason why we chose this cre line for our study. Since the RGC population is not homogeneous and include more than 40 different subtypes, we think that some of these subtypes do not support cre expression in these mice. The same reasoning can be applied to explain downregulation of Rbfox1 in amacrine cells, which are represented in the retina by more than 60 subtypes. The extent of Rbfox1 in amacrine cells was not quantified since the primary aim of the study was to evaluate the effect of Rbfox1 downregulation in RGCs and their survival in aged animals. We think that a complete Rbfox1 KO in 80% of RGCs was more than sufficient to address this aim. Furthermore, the remaining Rbfox1 positive RGCs in knockout animals can be viewed as internal controls.

Calbindin as a neuronal marker was used to show if there were any changes in the pattern of its expression in aged compared to young animal that we analyzed earlier (Plos1, 2018). Since calbindin is expressed not only in amacrine cells but also in other retinal neurons, including horizontal cells and in a relatively small population of RGCs, as correctly pointed by the reviewer, changes were made in the manuscript to clarify this and the reference to calbindin as a marker for amacrine cells in Fig. 2 legend was deleted. With respect to using amacrine cell marker, we have recently characterized the expression of Rbfox1 in various types of amacrine cells with several amacrine cell specific markers (Biosci Rep. 2022).

We have extensive experience working with different models of RGC degeneration and analysis of the extent of RGC loss/survival at different times post injury. The potential effect of Rbfox1 on cell survival in this study was evaluated at one week post injury, the time at which progressive RGC loss is still underway (~50%-60%). At two weeks post injury, when the most vulnerable cells are already lost (ONC leads to 80%-90% of RGCs loss by 2 weeks after injury), the effect on cell survival could be difficult to detect, especially if it is relatively modest.

Reviewer 2 Report

Gu et al. describes how aging Rbfox1 knockout mice develop visual function deficits by using visual cliff test, and down-regulation of Rbfox1 does not have protective effect on RGC survival in optic nerve crash model. Overall, the data that are presented are convincing and clearly displayed. However, these observations have been well-documented by the same authors in several publications. The lack of effort in addressing why visual function is compromised made this paper less appealing.

1. The most interesting data in this paper is the clear visual defect found in Rbfox1 knockout. This defect has been described in several previous publications, and some of the affected genes in retinas have been identified by the authors. However, in the retino-geniculo-cortical pathway outside the retina, how does Rbfox1 knockout affect other cells in different areas along this pathway and contribute to such visual defect have not been addressed. It is well known that Rbfox1 knockout in the brain developed spontaneous seizures and increased neuronal excitation. The UBC-CreErt2 used in this paper likely caused Rbfox1 deletion in other regions of the brain. How would these defects in the brain might have affected the visual behaviors?

2. The authors have previously identified several genes in younger Rbfox1 knockout retinas, how are those genes affected in the aging Rbflox1 retinas? If scRNA-seq or scSmart-seq could be performed, it would provide more insights.

3. In discussion, authors discussed a link between Rbfox1 and stress-response. I do not see any reference regarding how Rbflox1 regulates stress-response. Relevant to this, authors discussed the redundancy of Rbfox family genes in retinal neurons. Again, I do not see any effort to address whether these genes might have compensated the loss of Rbfox1.

Author Response

We thank the reviewer for the comments and helpful suggestions to improve our manuscript. All of the issues that were brought up were taken into consideration in the revised manuscript and addressed point-by-point below. We believe that the changes made to the manuscript significantly improved its quality.

The UBC-CreERT2 was used to achieve cre expression in RGCs, since, to our knowledge, there is no cre line that produce RGC-specific or retina-specific (including RGCs) cre expression. Cre expression in these mice is not ubiquitous, but it is not restricted to RGC. Therefore, we do not exclude the possibility that Rbfox1 downregulation take place in neurons within brain regions that are involved in processing of visual information relevant to depth perception. Unfortunately, Rbfox1 KO produces no cellular phenotype in our model (also in neuronal KO model) that would help to localize the region along the visual pathway that may be associated with this deficit of visual function. We are currently working on the generation of RGC-specific Rbfox1 KO, which will help us determine whether this behavioral deficit is associated with abnormal RGC function. With respect to the possible involvement of brain regions, we are in the process of analyzing Rbfox1 expression in LGN and V1 (although extrastriate visual cortex may also be involved) of Rbfox1 KOs. The potential role of Rbfox1 downregulation in the brain regions that may be associated with depth perception deficit is discussed in the revised manuscript.

We agree with the reviewer that RNAseq would provide more insight about age related differences in molecular composition of Rbfox1 KO retinas. However, since there is no effect of Rbfox1 downregulation on retinal phenotype and the number of RGCs in aged animals was observed, we do not expect significant alterations in the expression of Rbfox1-regulated genes in aged vs young animals’ retinas.

The following studies were referenced for regulation of Rbfox1 of stress-response pathways: Kucherenko, M.M.; Shcherbata, H.R. Stress-Dependent MiR-980 Regulation of Rbfox1/A2bp1 Promotes Ribonucleoprotein Granule Formation and Cell Survival. Nature Communications 2018.

el Fatimy et al. MicroRNA-132 Provides Neuroprotection for Tauopathies via Multiple Signaling Pathways. Acta Neuropathologica 2018.

A discussion about possible redundancy in the functions of Rbfox proteins that may explain a limited effect of downregulation of individual members of this family on cellular phenotype and function, has been added to the revised manuscript.

Round 2

Reviewer 1 Report

Overall the authors have not done enough to address all of the reviewer concerns.

Regarding use of Calbindin to label dACs. Although, not ideal, rewording this passage is sufficient. However, please adjust wording to "most dACs" in the text to reflect that there not all dACs are labeled by calbindin.

Regarding Robfox1 knockout in the front of the eye. It is very helpful to the reader to know that you have carefully examined normal function here. Please go further than just indicated what tests were used. What were your inclusion/exclusion criteria? If any of this was quantified, please indicate as such.

Regarding the Cre expression system, this is still not adequately addressed. Without knowing, which cells are driving Cre it is extremely difficult to interpret this dataset. In lieu of a different Cre driver, the authors at least need to cross their Ubq-CreER mice with an accepted ubiquitous reporter (e.g. Ai9) and carefully catalogue which cells are and are not induced. An alternative expression system would be preferable. For example AAV-Cre-GFP driven by RGC enriched promoter elements would be MUCH cleaner than the current system even if all RGCs are transfected, the untransfected RGCs can at least be identified and excluded. And while some promoter leakage would be expected, at least it won't be throughout the animal. This is even more relevant for a behavioral test which includes motor and 'fear' aspects like the visual cliff. Additionally, tamoxifen induction is often non-complete.

Regarding the time course of examination. The purpose of choosing a time point where death of RGCs is more stable is because when you examine in the stronger slope of the cell death curve, you're results are much more influenced by delays to cell death disguised as cell rescue. If RGCs are alive at 1 week, but not 2 weeks you haven't protected anything. Since no protection was reported in this study, perhaps it is of less consequence.

Author Response

The wording regarding AC labeling by calbindin has been modified as recommended.

The following Inclusion and Exclusion Criteria section has been added to the methods:

Twenty-two-month-old Rbfox1fl/fl/UBC-Cre+/- and age-matched heterozygous Rbfox1fl/+ control mice were included in this study. Prior to the ONC procedure, exclusion criteria were: visible corneal abrasion, opacity, inflammation or edema and cataract. A fundus examination was performed under an operating microscope immediately after the ONC procedure to exclude animals with blockage of retinal blood flow from the study. The exclusion criteria were also included: surgical complications including infection, bleeding, lens discoloration, or purulent drainage from the wound. No animal that entered the study was rejected due to the above-listed exclusion criteria.

We understand the reviewer's concern regarding the model and agree that downregulation of a target gene in specific cells makes it easy to evaluate its role in those cells. However, all constitutive and the vast majority of conditional KO animal models do not target specific cells with high efficiency. With respect to gene KO in RGCs, there are no available transgenes or AAVs that provide efficient and specific gene KO. There are a number of studies that try to address this issue. For instance, Drayson and Triplett (Genesis, 2019) showed that in the Cre BAC transgenic line, Cre expression is restricted to various types of RGCs in the retina and sparsely expressed in the brain, excluding retinorecipient regions. However, in Fig. 2C it can be seen that the majority of cells in the GCL (even considering that mouse GCl contains ~1:1 ratio of RGCs and dACs) are not targeted in this transgene. We are working on the identification of promoter regions that control RGC-specific expression of Rbpms that can be used to drive gene expression (including Cre) in all types of RGCs. This gene was identified as an RGC marker in our lab (IOVS 2010), and data describing its basic promoter were published (MGG, 2018). In the model that was used in our study, although Cre can be expressed in many different cell types, its effect will take place only in cells that coexpress both Cre and a floxed gene (in our case, it is Rbfox1), and in the retina, these cells are RGCs and ACs. We showed that Rbfox1 downregulation in more than 80% of RGCs and relatively small numbers of ACs has no effect on retinal morphology or on the survival of injured RGCs. The possibility that these results would be different if there was no potential Rbfox1 KO in the retinorecipient neurons in the brain is extremely unlikely. Using an efficient retinal specific or RGC specific Cre expression model, if there is one, may not exclude this possibility, unless anterograde transfer of Cre to RGC targets in the brain is ruled out. With respect to depth perception deficiency, Rbfox1 KO in the retina or brain regions that are involved in the retino-geniculo-cortical visual pathway may be responsible for this effect. Although we don’t know where this visual pathway is altered, our data show that Rbfox1 is involved in the processing of visual information. Also, if the location of this dysfunction is beyond the retina, we would miss it if the retinal or RGC specific KO model was used. Experiments to identify the regions responsible for this visual deficit were described in our first reply to reviewers’ comments (please see our response to Reviewer 2 comments). Regarding the identification of Cre expressing cells in the Ubq-CreER transgene, this information can be found on the JAX website describing these mice.

Since Rbfox1 downregulation had no effect on RGC survival at 1-week after ONC, evaluating its effect at later time points was unnecessary.